

# 1 Impact of Eastern and Central Pacific El Niño on Lower
# 2 Tropospheric Ozone in China

**Zhongjing Jiang[1], Jing Li[1]**
[1] Department of Atmospheric and Oceanic Sciences, School of Physics, Peking University,
Beijing, China
**Correspondence:** Jing Li ([jing-li@pku.edu.cn](mailto:jing-li@pku.edu.cn))
**Abstract**
Tropospheric ozone is an essential atmospheric component as it plays a significant role in
influencing radiation equilibrium and ecological health. It is affected not only by anthropogenic
activities but also by natural climate variabilities. Here we examine the tropospheric ozone change
in China associated with the Eastern Pacific (EP) and Central Pacific (CP) El Niño using satellite
observations from 2007 to 2017 and GEOS-Chem simulations from 1980 to 2017. GEOS-Chem
simulations reasonably reproduce the satellite-retrieved lower tropospheric ozone (LTO) changes
despite a slight underestimation. Results show that El Niño generally exerts negative impacts on
LTO concentration in China, except for southeastern China during the pre-CP El Niño autumn and
post-EP El Niño summer. The budget analysis further indicates that for both events, LTO changes
are dominated by the transport process controlled by circulation patterns and the chemical process
influenced by local meteorological anomalies associated with El Niño, especially the solar
radiation and relative humidity changes. The differences between EP and CP-induced LTO
changes mostly lie in southern China. The different strengths, positions, and duration of western
North Pacific anomalous anticyclone (WNPAC) induced by tropical warming are likely



responsible for the different EP and CP LTO changes. During the post-EP El Niño summer, the
Indian ocean capacitor also plays an important role in controlling LTO changes over southern
China.
**Key Words**
Lower tropospheric ozone, El Niño, meteorological fields, WNPAC, GEOS-Chem
## 1. Introduction
Tropospheric ozone is an important greenhouse gas and a major air pollutant affecting human
health and the ecosystem (Fleming et al., 2018; Maji et al., 2019; Mills et al., 2018). Produced
from the photochemical oxidation of carbon monoxide (CO) and volatile organic compounds
(VOCs) in the presence of nitrogen oxides (NOx) and sunlight, tropospheric ozone concentration
is largely affected by meteorological conditions, including solar radiation, relative humidity,
temperature, etc., which can influence the precursor emissions and photochemical reaction rates
(Guenther et al., 2012; Jeong et al., 2018). Thus, El Niño-Southern Oscillation (ENSO), as one of
the most prominent interannual climate variabilities, can influence ozone concentration by
affecting the local meteorological fields and modulating the ozone distribution through changes in
atmospheric circulation (Bjerknes, 1969; Chandra et al., 1998; Oman et al., 2013; Sudo and
Takahashi, 2001).

Because ENSO is a tropical signal, the majority of previous studies focus on discussing the impacts
of ENSO on tropical tropospheric ozone (Oman et al., 2011; Ziemke et al., 2010; Ziemke and
Chandra, 2003). A few studies demonstrated that the influence of ENSO on tropospheric ozone


could also extend to subtropics and mid-latitudes. Over Southeast Asia, Marlier et al., (2013) show
that during the strong El Niño years, fires contribute up to 50 ppbv in annual average ozone surface
concentrations near fire sources. Over the US, Xu et al., (2017) examined the impact of ENSO on
ozone during 1993 to 2013 and found that the monthly ozone decreased about 1.8 ppbv per
standard deviation of Niño 3.4 index during El Niño years. They found significant spatial
dependence and seasonality of ENSO's influence on ozone. ENSO affects surface ozone via
different processes during warm or cold seasons in different regions in the US. As for China, a few
studies discussed the impact of ENSO on the total column or tropospheric column ozone
concentration over part of China, such as Tibet, or include China as part of their study regions
(Koumoutsaris et al., 2008; Singh et al., 2002; Xu et al., 2018; Zou et al., 2001). However, studies
that specially focused on the influence of ENSO on tropospheric ozone over China are still limited.
Yet, ENSO may exert profound impacts on the temperature, precipitation in China both in its
developing season and the following year (Cao et al., 2017; Fang et al., 2021; Li et al., 2021, 2018;
Xu et al., 2018) and then affect ozone concentrations. In view of the severe ozone pollution in
China and the substantial role of natural impacts, it is essential to clarify how ozone concentrations
in China respond to ENSO.

On the other hand, increasing studies have noticed the different flavors of ENSO. A widely
accepted view is to categorize El Niño into the Eastern Pacific (EP) and Central Pacific (CP) El
Niño (Ashok et al., 2007; Yeh et al., 2009), whose positive sea surface temperature (SST)
anomalies are located over the eastern and central Pacific respectively. Whereas previous studies
about the response of ozone to ENSO generally used Niño 3.4 index (Olsen et al., 2016; Oman et
al., 2013) or the Southern Oscillation index (Koumoutsaris et al., 2008; Ziemke and Chandra,





2003) to represent the intensity of ENSO and the difference between the two types of El Niño is
neglected. Yet, due to the different generation mechanisms (Yu et al., 2010). the two types of El
Niño can induce various changes in climate or synoptic weather in the tropics as well as the mid-
to-high latitudes (Shi & Qian, 2018; Yu et al., 2012). Studies have shown that different types of
El Niño can induce different changes in tropical cyclone genesis, water vapor transport, and rainfall
patterns over China (Feng et al., 2011; Li et al., 2014; Wang and Wang, 2013). As such, they also
likely exert different impacts on atmospheric components. Some research explores the
teleconnections of different types of El Niño with climate anomalies and haze pollution in China
(Gao et al., 2020; Ren et al., 2018; Xu et al., 2018; Yu et al., 2019, 2020), whereas few studies
have discussed the teleconnections between ozone and different El Niño types, which is thus the
focus of this study.

In this paper, we investigate the changes of tropospheric ozone in China associated with EP and
CP El Niño, using satellite observations and the GEOS-Chem chemistry transport model
simulations. This study aims to explore how El Niño influences the lower tropospheric ozone in
China to shed light on the ozone air quality control on the interannual timescale. In addition, we
hope this study can also improve our understanding of the mechanism of teleconnections between
ENSO and tropospheric ozone concentration in mid-latitudes.

**2. Data and Methods**
**2.1 The classification of Eastern and Central Pacific El Niño**
To distinguish the type of El Niño, we first used the Oceanic Niño Index (ONI) from the Climate
Prediction Center (CPC) of the National Oceanic and Atmospheric Administration (NOAA) to



filter out El Niño events. The ONI is defined as the 3-month running mean of ERSST.v5 SST
anomalies in the Niño 3.4 region (5°N-5°S, 120°W-170°W), based on centered 30-year base
periods updated every five years. An El Niño event is defined as when ONI is greater than or equal
to 0.5℃ for a period of at least five consecutive overlapping seasons. Then we combined two
methods, namely the Niño3/4 method in (Yeh et al., 2009) and the ENSO Modoki index (EMI)
method in (Ashok et al., 2007), to discriminate EP and CP El Niño. When the two methods show
consensus results, we define it as a typical EP or CP event.

**Niño3/4 method**
We first adopted the same Nino3/4 method in (Yeh et al., 2009). The classification is based on the
comparison between boreal winter (DJF) seasonal mean Niño 3 and Niño 4 indices. DJF Niño3
SST index is defined as the DJF seasonal SST anomaly in Niño 3 region (150°W-90°W, 5°N-5°S),
and DJF Niño 4 SST index is defined as the DJF seasonal SST anomaly in Niño 4 region (160°E-
150°W, 5°N-5°S).  The first step is to pick out the years when the DJF Niño3 and Niño 4 indices
both greater than 0.5℃,  then make the comparison between DJF Niño 3 and Niño 4 SST indices.
When DJF Niño3 SST index is greater than DJF Niño4 SST index, it is defined as an EP El Niño
event, otherwise as a CP El Niño event.

**El Niño Modoki index (EMI) method**
Ashok et al., (2007) derived an El Niño Modoki index (EMI) to capture whether there is a typical
CP-type event.
$$EMI = [SSTA]_A - 0.5 \times [SSTA]_B - 0.5 \times [SSTA]_C$$



[SSTA]$_A$, [SSTA]$_B$, and [SSTA]$_C$ represent the area-averaged SST anomaly of region A (165°E-
140°W, 10°S-10°N), B (110°W-70°W, 15°S-5°N), and C (125°E-145°E, 10°S-20°N) respectively.
We call a CP El Niño event "typical" when the index amplitude is equal to or greater than 0.7σ,
where σ is the seasonal standard deviation.

The classification results of EP and CP El Niño of the total 12 events from 1980-2017 are shown
in Table 1.

**2.2 Satellite-retrieved ozone and meteorological data**
Ozone abundance in the atmosphere can be measured from space using different remoting-sensing
techniques. Frequently used tropospheric column ozone datasets include OMI/MLS carried by
AURA and Infrared Atmospheric Sounding Interferometer (IASI) carried by the MetOp satellites.
As we prefer to consider lower tropospheric ozone in this study, we choose to use IASI, which can
retrieve the ozone from the surface to 6 km. In addition, IASI is also a superior choice considering
the spatial coverage, resolution, and data quality. IASI is a thermal infrared Fourier transform
spectrometer onboard the MetOp-A and B satellites; as a space-borne nadir-viewing instrument, it
probes the troposphere using the thermal infrared spectral range, and the atmospheric data is
further retrieved by inversion algorithms (Boynard et al., 2009, 2016). The IASI-A and B
instruments have been operationally providing atmospheric products since October 2007 and
March    2013,    respectively.    Ozone    monthly    gridded    data    is    available    on
https://cds.climate.copernicus.eu/cdsapp#!/dataset/satellite-ozone-v1?tab=form,    last    access:   8
November 2021. We used the ozone data from September 2007 to Autumn 2017, mostly from
MetOp-A v0001, with substitutes from MetOp-B v0001 for several missing months in 2015.




Meteorological fields for 1980-2017 were obtained from the Goddard Earth Observing System
(MERRA-2) database (Bosilovich et al., 2016), which is the current operational met data product
from the Global Modeling and Assimilation Office (GMAO). The data are available at
http://ftp.as.harvard.edu/gcgrid/data/GEOS_2x2.5/MERRA2/, last access: 8 November 2021.
Meteorological variables used in section 3.2 include surface downwelling solar radiation (SR),
relative humidity (RH), total precipitation (TP), temperature (T), sea level pressure (SLP), and
wind fields. RH, T, and winds are multi-level variables. We calculated the 0-6 km column averages
to be consistent with ozone, whereas SR, TP, and SLP are single-level variables.

**2.3 GEOS-Chem simulations**
The GEOS-Chem (GC) chemical transport model (Bey et al., 2001; v12.3.2; http://geos-chem.org)
is used to explore the EP and CP El Niño-related tropospheric ozone changes. We choose the
standard chemistry mechanism, which includes both troposphere and stratosphere. The Universal
tropospheric-stratospheric Chemistry eXtension (UCX) mechanism developed by Eastham et al.,
(2014) combines both tropospheric and stratospheric reactions into a single chemistry mechanism.
The model is driven by MERRA-2 meteorological fields with 72 vertical levels and 2°× 2.5°
horizontal resolution. We first performed a historical run from 1980-2017 with anthropogenic
emissions fixed at the year 2000, so the difference among different events is only caused by the
meteorological fields. A drawback of this setting is that the biomass burning is also fixed at the
year 2000; however, the biogenic emission will still change as it was calculated interactively
with meteorology.



The simulated ozone concentration is further validated against tropospheric ozone within the same
altitude range retrieved by the IASI. Because IASI only retrieves column ozone concentration
between 0-6 km, our comparison and analysis also focus on 0-6 km integrated column ozone
concentration, referred to as lower tropospheric ozone (LTO) thereafter. As satellite observation
starts in October 2007, to ensure comparability, we selected the 2015-2016 and 2009-2010 events
to represent EP and CP El Niño, respectively. A 10-year average (September 2007-Autumn 2017)
was used as the climatological state. Figure S1 shows the seasonal mean SST anomalies for the
two periods selected, which corresponds well to EP (2015-2016) and CP (2009-2010) El Niño
patterns.

To further distinguish the ozone changes between EP and CP El Niño, we also performed three
composite model simulations driven by the composite meteorological fields for the four seasons
of three most typical EP events (1982-1983, 1997-1998, 2015-2016), four CP events (1994-1995,
2002-2003, 2004-2005, 2009-2010), and a 30-year averaged climatology (September 1985-
Autumn 2015) for comparison. Figure S2 shows the composites of seasonal mean SST anomalies,
which well corresponded to EP and CP El Niño.

Moreover, to explain the physical and chemical drivers of the ozone changes, we analyzed the
composite meteorological fields to check the ENSO-related meteorological changes. We also
diagnosed the 0-6 km ozone budget changes of different model processes and quantified the
absolute contribution of each process. These budget diagnoses are calculated by taking the
difference in 0-6 km vertically integrated column ozone mass before and after major GEOS-Chem
simulation components, including chemistry, transport, mixing, and convection, at each timestep.




### 3. Results

### 3.1 Lower tropospheric ozone changes associated with EP and CP El Niño

An ENSO event usually develops in autumn ($SON_0$), reaches its peak in winter ($DJF_{0-1}$), and
decays in the following spring ($MAM_1$) and summer ($JJA_1$) (Xu et al., 2017). We denote the ENSO
developing year as year 0 and the following year as year 1. We first compare the climatology state
(Figure S3) between observation and simulation. Model performance is comparable to those in
previous modeling works (Dang et al., 2021; Lu et al., 2019; Ni et al., 2018). The bias mainly
comes from the resolution, chemical mechanism, microphysics processes, and site
representativeness(Sun et al., 2019; Young et al., 2018). Then we examine the change of satellite-
retrieved and simulated 0-6 km column ozone during the 2015-2016 EP and 2009-2010 CP events
with respect to climatology (Figure 1) to validate the model response to ENSO-related signals.

EP El Niño generally exerts negative effects on LTO in China in both observation and simulation,
except for a dipole mode change over southern China during pre-EP autumn and post-EP summer.
The satellite-retrieved LTO shows an increase in the south and a decrease in the north in autumn,
while this dipole mode is obscure in the simulation. In winter and spring, both the satellite-
retrieved and simulated LTO exhibit coherent decreases over the whole of China, but the intensity
in the model is much smaller. In summer, the observation still shows declines over most regions
except a slight increase over the southeast coastal area and southwestern China. The simulation
shows a similar pattern but with much stronger positive signals over southern China. In contrast,
in CP El Niño there are more prominent LTO increases, such as over southern China in autumn,



over northeastern China in spring, and over northern China in summer. In autumn, the satellite
observation and simulation both exhibit a dipole mode change in the north and south with LTO
decrease over northern and increase over southern China. In winter, the observed and simulated
LTO shows a reverse change with slightly positive and negative signals. The LTO changes in
spring and summer are pretty consistent between observation and simulation.

In general, the LTO changes are at -1~1 DU (Figure S4), accounting for 5~10% of the 0~25 DU
mean range. The spatial patterns of the simulated and observed LTO changes agree well, despite
an overall underestimation by the model. This underestimation can be explained by the fixed
biomass burning emission in the simulation that weakens the sensitivity of tropospheric ozone to
ENSO, as this leads to milder changes in ozone precursors such as carbon monoxide. The
underestimation in spring and summer is most significant at high latitude areas, such as
northeastern China, for both EP and CP events. This deviation probably represents the
interferences of other high latitude climate variabilities. Another reason is that the model
underestimates the average ozone concentration at high latitudes in winter and spring (Figure S3),
probably due to the unprecise halogen chemistry (Wang et al., 2021) and the poor represent of
Brewer-Dobson circulation in the model. Thus, the ozone transport from polar regions to northern
China can be much less in the model. The overall consistency between simulated and observed
LTO changes gives us the confidence to use the model for composite analysis, as the satellite
record only covers limited El Niño events.

To include more El Niño events and check the response of ozone to meteorological fields, we
further used the composite meteorological fields of three EP events and four CP events to drive





the GEOS-Chem model. Figure 2 shows the LTO changes in China during different seasons of the
EP and CP El Niño. It is seen that LTO decreases over most regions in both EP and CP type in the
range of 5~10%, whereas only some regional increases are seen in pre- El Niño autumn and post-
El Niño summer. LTO decrease consistently during winter and spring, reaching ~10% for western
and northern China. It appears that the seasonal alternation of LTO changes in southern China may
represent the extension of the remarkable ozone changes over the tropical regions. During the EP
(CP) El Niño developing, sustaining, and first decaying periods, there are significant dipolar
(tripolar) modes of ozone changes over the tropical Pacific area (Figure S5), which is consistent
with the result of previous studies (Chandra et al., 1998; Oman et al., 2013). These ozone changing
patterns correspond well with solar radiation changes (Figure S6) since they can modulate the
photolysis rates and biogenic emissions. The enhancement of tropospheric ozone production over
the west Pacific retreats to lower latitudes in winter and spring when the sun moves to the southern
hemisphere; therefore, LTO coherently decreases in China. During winter and spring, the changes
associated with CP El Niño are more extensive, spatially uniform, and stronger than EP. For
summer, however, EP appears to associate with a more substantial LTO decrease, especially for
the northern and southwestern parts. For summer, however, EP appears to associate with a more
substantial LTO decrease, especially for the northern and southwestern parts. The region
exhibiting the most LTO change differences between EP and CP events is southern China. The
differences between EP and CP patterns will be further examined in the next section.

Because El Niño is generally associated with decreased tropospheric ozone concentration, we also
briefly examine the LTO changes during the negative phase, i.e., La Niña events (Figure S7). In
contrast to El Niño, La Niña tend to be associated with extensive LTO increases by ~2-5%,



especially over northern China, indicating an adverse impact on the already severe tropospheric
ozone pollution in this region. An increase in ozone concentration during the post-La Niña spring
has also been reported by (Wie et al., 2021). However, because El Niño teleconnections are
typically stronger and better established, we still focus on El Niño in this study.

**3.2 Differences in ozone changes associated with EP and CP El Niño**
To clarify the mechanism associated with different LTO changes of the two types of El Niño, we
further examine the changes of meteorological variables, including SR, RH, TP, T, SLP, and wind
fields during EP (Figure 3) and CP events (Figure 4). The leading two variables impact the local
production, and the circulation changes represented by SLP and winds control the regional
transport. Although wet scavenging of ozone by TP is insufficient because ozone is insoluble in
water, TP is closely related to SR and RH; it is also the primary variable examined to identify
ENSO teleconnection. We thus also include TP in the comparison. In addition, we calculate the
budget changes corresponding to the EP and CP events from GEOS-Chem simulations. Because
chemistry and transport are the two dominant processes accounting for more than 70% of the ozone
changes in all conditions (Figure 5), we focus our following discussions on these two processes
(Figure 6).

In the autumn before El Niño, LTO changes for EP type show a general decrease in China (Figure
2a), especially in the southeastern part. EP El Niño is always accompanied by an anomalous
anticyclone in the Philippine sea (Figure 3q), which produces strong southwestern wind anomalies
that transport moisture from the ocean, resulting in increased TP and RH but decreased SR over
southeastern China (Figure 3i, e, a). These changes are unfavorable for ozone production but



efficient for ozone removal, thus leading to a chemical loss of LTO over southern China (Figure
6a). Some regional increase over southwestern China is observed and likely due to the positive
contribution of transport (Figure 6e) from India as indicated by the west wind anomalies (Figure
3q). During the CP event, there is a moderate dipole mode change (Figure 2e), with decreases in
northern China and increases in the southern part. In contrast to EP, an anomalous cyclone appears
over the Philippine sea, leading to northwest wind anomalies over southern China that produces a
dry condition with increased SR (Figure 4i, e, a). The slight decrease in LTO over northern China
is likely attributed to the decreased chemical production (Figure 6i) associated with negative
temperature anomalies (Figure 4m), although the signal is not statistically significant. The opposite
atmospheric circulation patterns over the Philippian sea during EP and CP events are responses to
the different SST anomaly regions under these two conditions, as shown by (Wang and Wang,
2013) using simple atmospheric model experiments.

In winter, when the Pacific SST anomalies reach their maxima, EP and CP El Niño are both
associated with increased TP, RH, and decreased SR over southern China (Figure3b,f,j & 4b,f,j).
These similar changes are due to the moisture transport induced by western North Pacific
anomalous anticyclones (WNPAC) that occur in both EP and CP El Niño, while EP exhibits greater
meteorological changes than CP due to the much stronger anomalous anticyclone (Figure 3r& 4r).
As a critical system that links El Niño and East Asia climate change, WNPAC is initiated and
maintained by local atmosphere-ocean interaction (Wang et al., 2000) and the moist enthalpy
advection/Rossby wave modulation (Wu et al., 2017a, 2017b), the formation and maintenance
mechanisms are discussed thoroughly in (Li et al., 2017). Although the meteorological variables
change in the same direction, the EP and CP-related LTO changes in winter are still opposite over



southern China (Figure 2b&f), where the El Niño teleconnection signal is the most prominent
(Wang et al., 2020). Budget analysis reveals that this phenomenon is due to the varying
contribution of different model processes. Consistent with the increased RH and decreased SR, the
contributions of chemical processes are both negative over this region during EP and CP (Figure
6b&j). The southwestern wind anomalies (Figure 3r&4r) bring not only water vapor from the
ocean but also ozone from India and China-Indochina Peninsula to southern China, contributing
to LTO concentration there. During EP, the chemical loss (Figure 6b) exceeds the positive
transport (Figure 6f) due to the severe change of SR and RH over southern China (Figure 3b&f).
However, for CP conditions, the chemical loss (Figure 6j) due to the increased RH is much weaker
and is offset or even exceeded by transport (Figure 6n). This is also consistent with the much larger
absolute contribution of transport than chemistry for CP (Figure 5f).

In spring, LTO decreases extensively over entire northern China under both EP and CP conditions
(Figure 2c&g), coherent with the large-scale reduction of SR and increase of RH (Figure 3c,g
&4c,g). WNPAC maintains under EP conditions according to the moist enthalpy advection
mechanism (Wu et al., 2017a), whereas it nearly disappears in CP (Feng et al., 2011). In EP
condition, with the slight westward shift of the anticyclone center from winter to spring, the wind
anomalies also shift from southwesterly to southernly, bringing more moisture, and further
enhancing TP in higher latitudes where RH increases and SR decreases coherently. However, the
chemistry still contributes positively over eastern China (Figure 6c), which might be attributed to
the increased temperature related to the warm south winds (Figure 3o&s). As the climate warms
from winter to spring, the role of temperature becomes increasingly important. However, as the
southerlies blow low ozone air from the ocean, the severe negative transport (Figure 6g) dominates



the overall ozone decrease. In CP, however, regional transport is weaker due to the unremarkable
change of circulation patterns over the western north Pacific compared to the EP condition; thus,
the absolute contribution of transport and chemistry are comparable for CP (Figure 5g).

The situation for the post-El Niño summer is more complicated as El Niño teleconnections
substantially involve air-sea interactions and inter-basin teleconnections (Feng et al., 2011; Kug et
al., 2009). Ozone changes for the EP condition show a decrease over central and northern China
and a band-like ozone increase over southeastern China (Figure 2d). Although the chemical
production (Figure 6d) increases with the slight SR increase and RH decrease (Figure 3d&h) over
China's eastern coastal line, the transport process (Figure 6h) controlled by southwestern wind
anomalies dominates the ozone decline over the Yangtze river basin and increases over the
southeastern coastal line. The circulation anomalies manifest as a tripolar pattern with an
anomalous anti-cyclone (AAC) over the southern China sea and an anomalous cyclone circulation
(ACC) over Japan (Figure 3t). This pattern appears to be induced by the Indian Ocean capacitor
(IOC) effect, which indicates the Indian Ocean (IO) memory of ENSO influence (Chen et al., 2012;
Xie et al., 2009; Yang et al., 2007). Since the convection is suppressed in the AAC, the drier
condition corresponds well with the positive LTO changes over the Philippine sea (Figure S5d).
This positive signal extends to southeastern China coastal areas due to the transport by the
southwest wind anomalies. During CP, ozone decreases coherently over most of China (Figure
2h). As no significant IO warming appears (Figure S2h), the summer climate is influenced more
by the western Pacific warm pool. The negative SST anomalies in the central-east Pacific imply
an upcoming La Niña. Accordingly, the western Pacific warm pool begins to shrink with the
building of La Niña (Johnson and Birnbaum, 2017). Associated with the SST drop, SLP increases



over the northwestern Pacific (Figure 4t), resulting in an enhanced western pacific subtropical high
(WPSH), which is a typical feature of CP El Niño (Chen et al., 2019). Controlled more by local
Pacific than IO, the SLP center shifts eastward compared to AAC in EP, and the positive LTO
anomalies also move eastward accordingly (Figure S5h). Regional transport (Figure 6g) by the
southwest wind anomalies surrounding the positive SLP center exerts a consistent negative
contribution to LTO in southern China (Figure 2h; Jiang et al., 2021). In sum, the post-El Niño
summer LTO change is dominated by the IOC effect for EP and WPSH enhancement for CP.

## 4. Conclusions and discussions

This study investigates the changes in tropospheric ozone concentration in China associated with
the EP and CP El Niño using satellite observations and GEOS-Chem chemical transport model
simulations. The general consistency between observed and simulated results confirms the model's
credibility. Overall, both types of El Niño exert a negative effect on LTO by 5~10%, with some
regional increases. The ozone changes were explained from the perspective of El Niño-induced
meteorological fields, which further lead to changes in local production, regional transport, etc.
Budget analysis indicates transport controlled by circulation patterns plays the leading role, and
chemistry affected by SR and RH plays the second role in driving the ozone changes. The
difference between EP and CP mainly lies in southern China. During the autumn, LTO decreases
(increases) about 4~8% (+2~4%) over southern China for EP (CP) type, corresponding well to
reversed changes of TP and related variables controlled by the different locations of SST
anomalies. In winter, the formed WNPAC maintains during both EP and CP, exerting a
counteracting effect on local production and regional transport. The impact of chemistry outweighs
the transport for EP, resulting in a slight LTO decrease over southern China (4~6%), vice versa



365 for CP (+0~2%). In spring, the WNPAC persisted under EP condition keeps exerting on ozone and

366 the transport dominates the overall decline of LTO for 5~10%, as the WNPAC disappeared in CP,

367 the role of transport weakens and the drier environment contributes to local production, which

368 leads to a slight ozone increase (+0~4%) over southern China. As for summer, the LTO decreases

369 5~10% in both types except for an increase over the southeastern coastal line for EP. Ozone

370 changes in EP type are dominated by the Indian ocean capacitor, and CP type are influenced more

371 by WPSH.

372

373 Our study indicates that natural variability, such as ENSO, can significantly impact lower

374 tropospheric ozone in mid to high latitudes. This has particular implications for ozone pollution

375 control in China. As much efforts have been taken to control anthropogenic emissions,

376 meteorological factors may play an increasingly important role in the future. The occurrence of El

377 Niño events produces a favorable environment for ozone pollution control in general, but caution

378 needs to be taken for southern China during CP autumn and EP summer. By contrast, when a La

379 Niña is predicted to occur, more strict emission control measures should be taken in the following

380 seasons, especially for northern China. Furthermore, by exploring the association between

381 different ENSO flavors and lower tropospheric ozone in China, this study enriches the theory of

382 ENSO teleconnection in mid-latitudes.

383

384 Nonetheless, there are still limitations in the current study that are subject to future improvements.

385 Tropospheric ozone concentration is influenced by stratospheric-tropospheric exchange (STE)

386 (Ding and Wang, 2006; Langford, 1999), although the effect is primarily concentrated in the upper

387 troposphere(Lin et al., 2015; Neu et al., 2014). Future work is needed to explain the difference of



388 ozone concentration in the vertical dimension and quantify the role of STE in the ENSO-induced

389 LTO changes. The role of biomass burning emission, which also varies with ENSO, will also be

390 investigated. Furthermore, long-term observations, especially in China, are needed to verify the

391 model results reported here.

392

393

394 **Code and data availability.**  The IASI satellite tropospheric column ozone data are available on

395 https://cds.climate.copernicus.eu/cdsapp#!/dataset/satellite-ozone-v1?tab=form,

396 doi:10.24381/cds.4ebfe4eb, last access: 8 November 2021. The MERRA2 meteorology data is

397 available    at    http://ftp.as.harvard.edu/gcgrid/data/GEOS_2x2.5/MERRA2/,    last    access:    8

398 November 2021 (Bosilovich et al., 2016). The GEOS-Chem model is a community model and is

399 freely    available    (http://wiki.seas.harvard.edu/geos-chem/index.php/GEOS-Chem_12#12.3.2,

400 doi:10.5281/zenodo.2658178, Yantosca, 2019).



403 **Author contributions.** JL and ZJ designed the study. ZJ ran the GEOS-Chem model and

404 performed the analysis. ZJ and JL wrote the paper.


406 **Competing interests.** The authors declare that they have no conflict of interest.


408 **Acknowledgments.** We appreciate GMAO for providing the MERRA-2 meteorological data. We

409 thank ECMWF for providing the ozone monthly gridded data. We also acknowledge the efforts of

410 the GEOS-Chem Working Groups and Support Team for developing and maintaining the GEOS-

411 Chem model.

412

413 **Financial support.**  This study is funded by the National Natural Science Foundation of China

414 (NSFC) Grant No. 41975023.

415



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

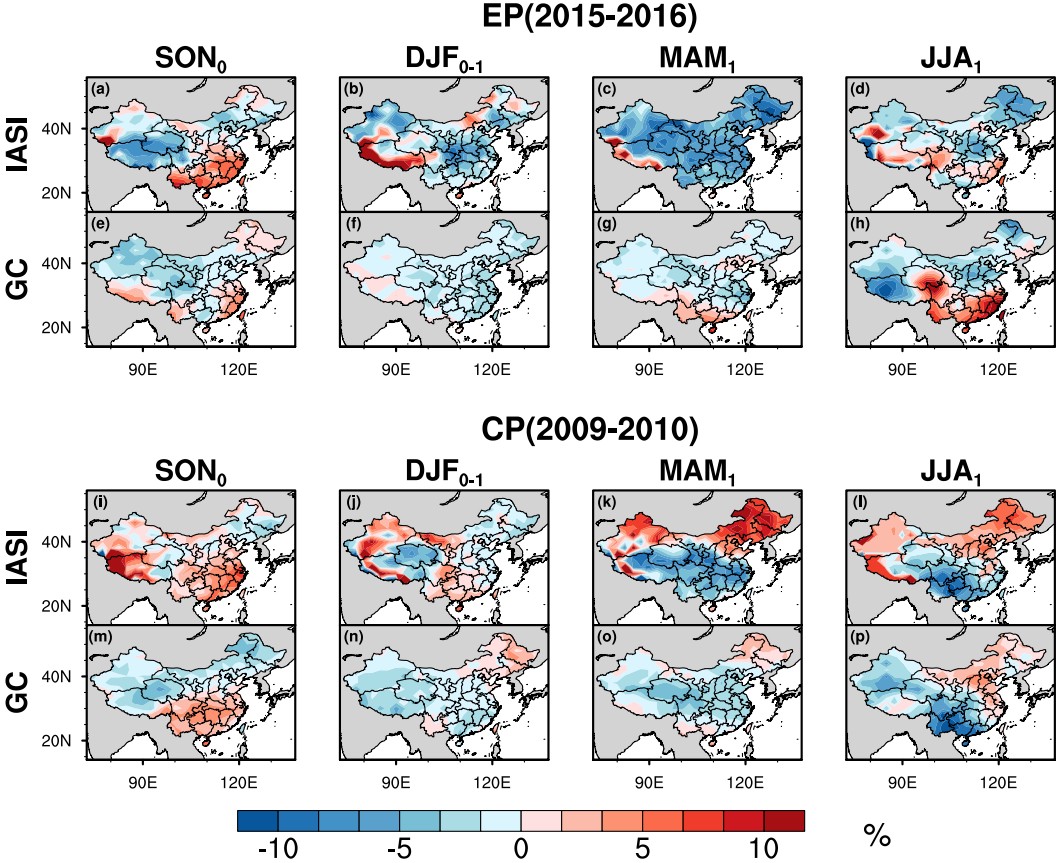

**Figure 1.** The percentage changes (unit: %) of satellite-observed (IASI) and model-simulated (GC) tropospheric column ozone (0-6 km, unit: DU) for four seasons in EP (2015-2016) and CP (2009-2010) El Niño years.

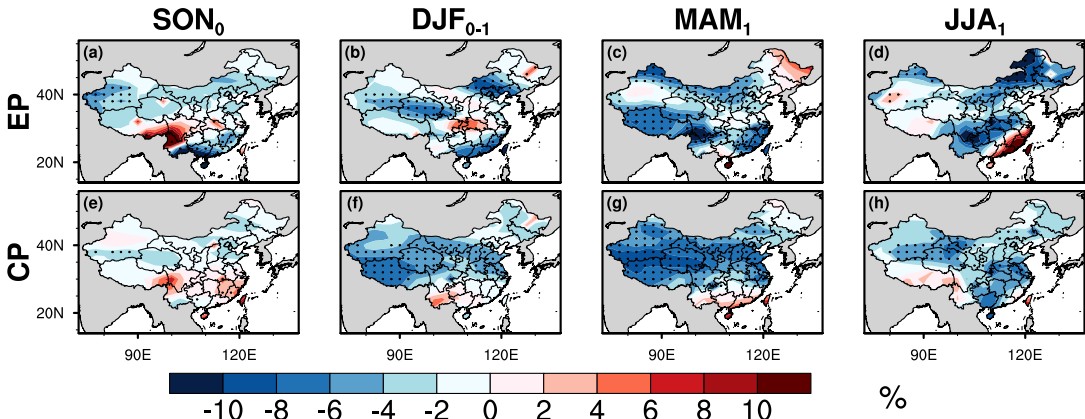

**Figure 2.** The percentage changes (unit: %) of simulated (GC) tropospheric column ozone (0-6 km, unit: DU) anomalies driven by composite meteorological fields for four seasons in EP and CP El Niño years. Black dots represent the 95% confidence level by t-test.

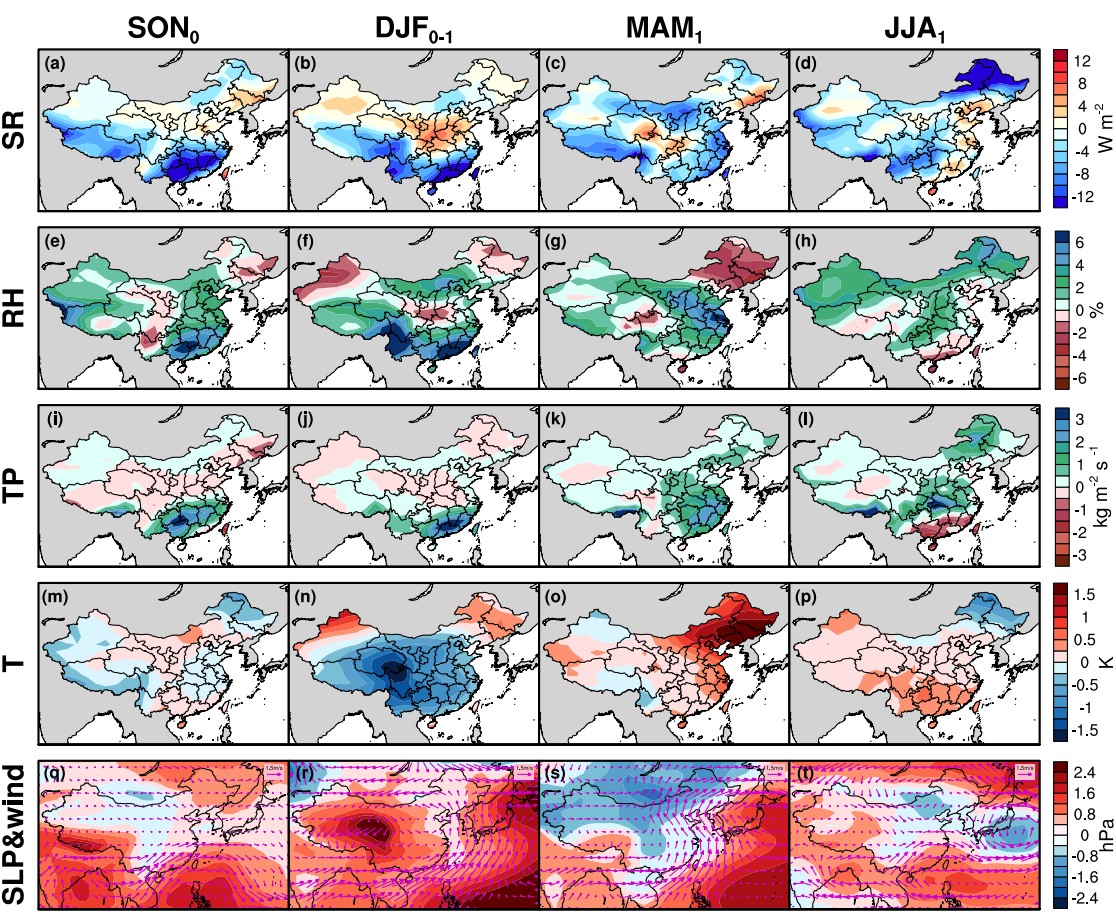

**Figure 3.** The composite anomalies of meteorological variables, including surface downwelling solar radiation (SR), relative humidity (RH), total precipitation (TP), temperature (T), sea level pressure (SLP), and winds, for four seasons in EP El Niño years.



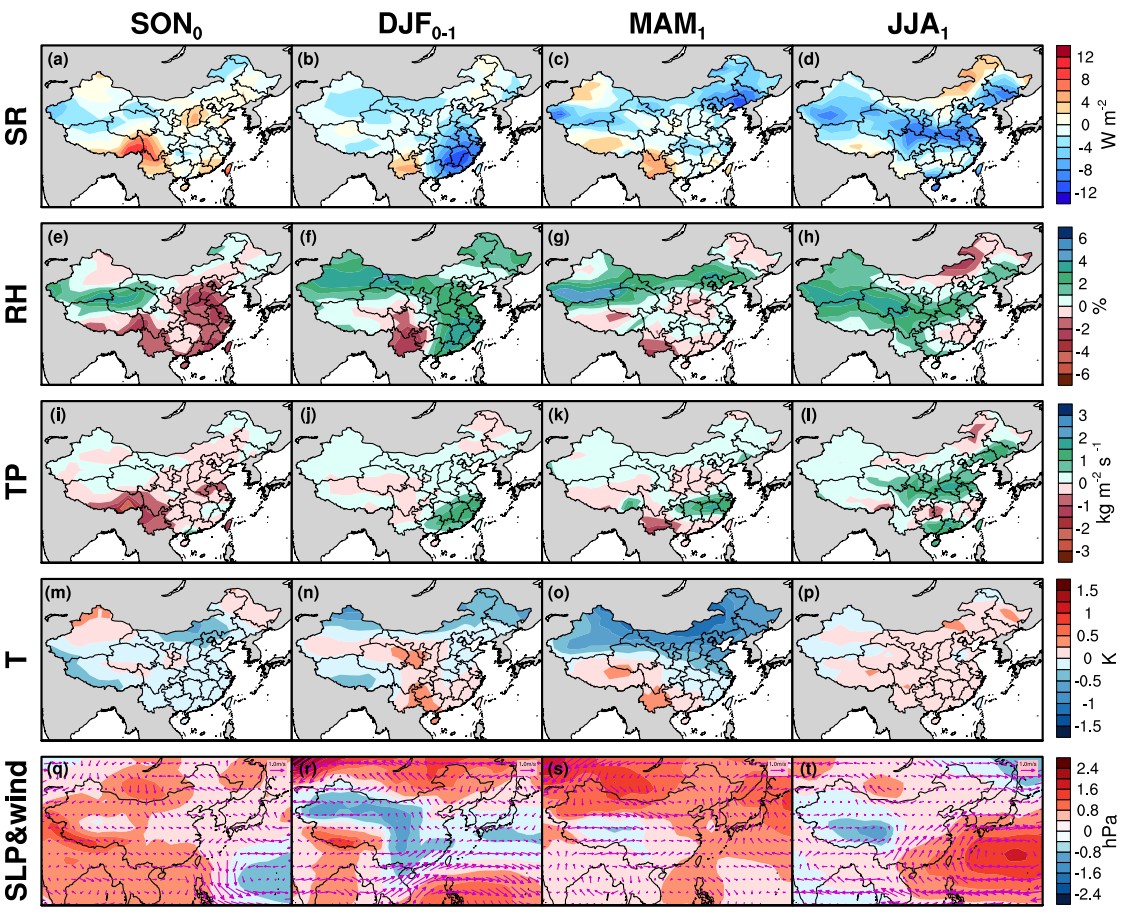

**Figure 4.** Same as Figure 3 except for CP El Niño.

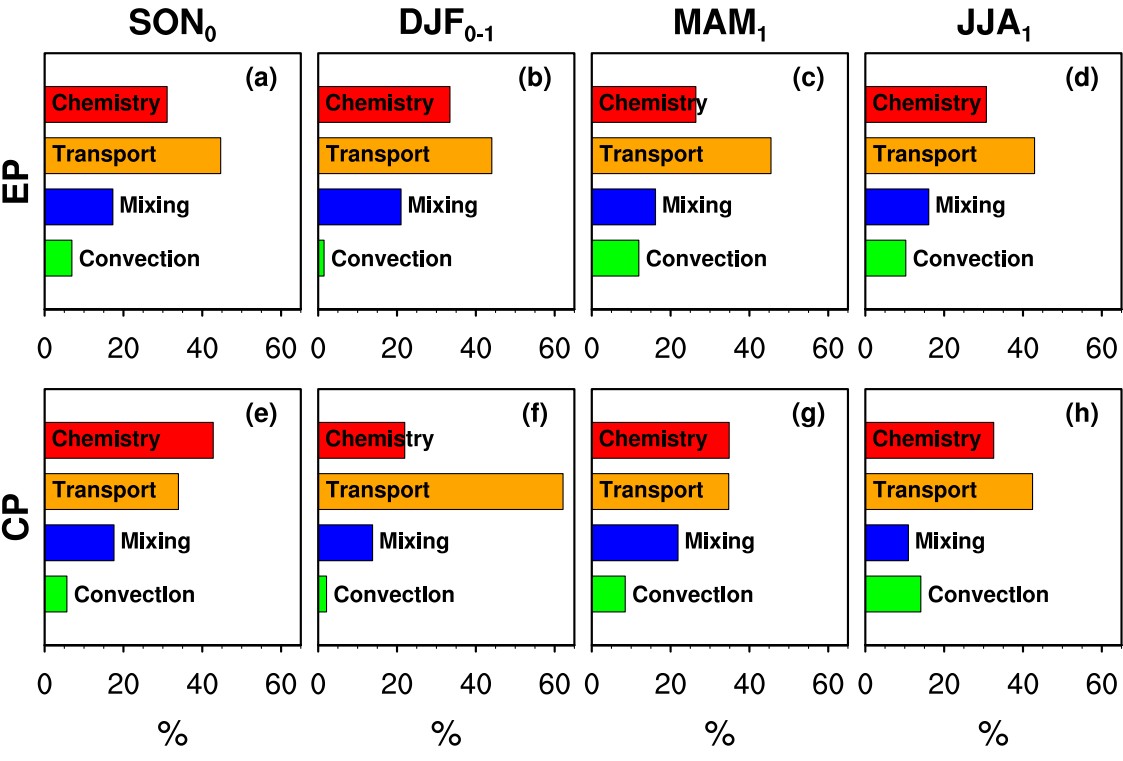

**Figure 5.** The absolute contribution (unit: %) of model processes, including chemistry, transport, mixing, and convection driven by the composite meteorological fields for four seasons in EP and CP El Niño years. These are the area-averaged values eastern China region (24.0–42.0°N, 100.0–117.5°E, purple box in Figure 6a).

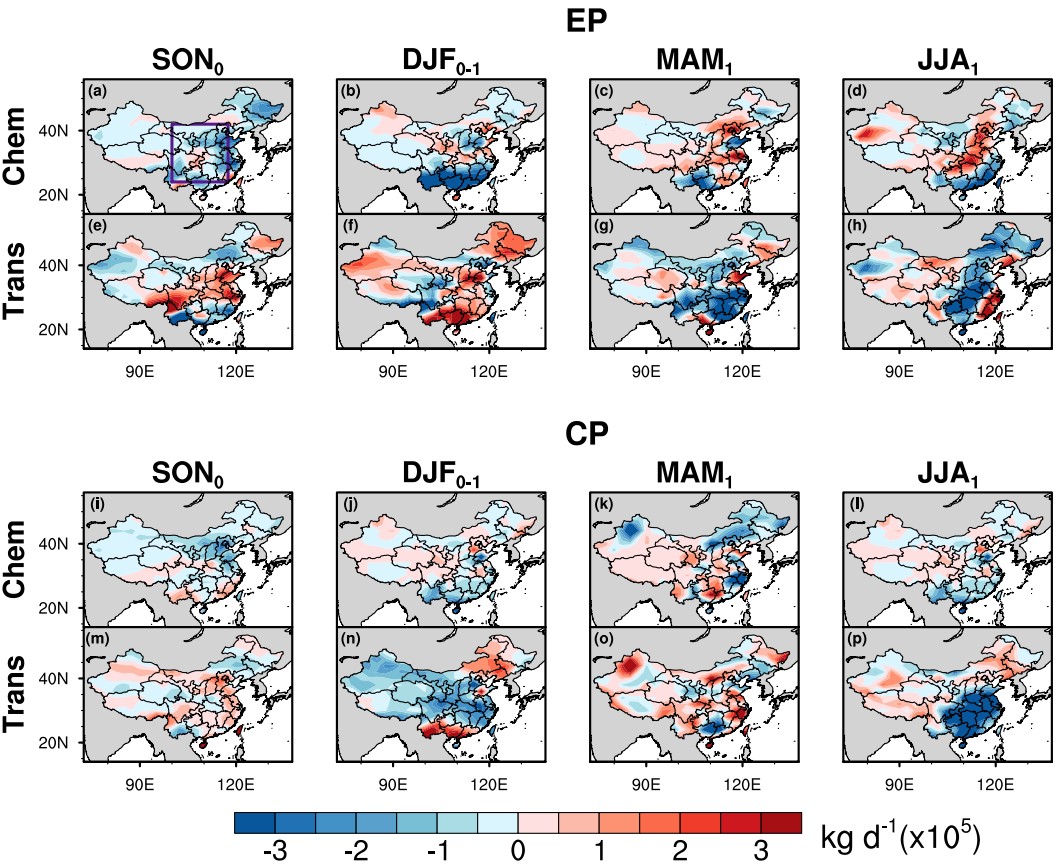

**Figure 6.** The composite tropospheric column ozone mass anomalies of chemistry and transport processes (0-6km, unit: kg d$^{-1}$) for four seasons in EP and CP El Niño years.





| ONI-El Niño year | type | | |
|---|---|---|---|
| | **Niño3/4 method** | **EMI method** | **consensus** |
| 1982-1983 | EP | EP | EP |
| 1986-1987 | / | EP | / |
| 1987-1988 | CP | EP | / |
| 1991-1992 | EP | CP | / |
| 1994-1995 | CP | CP | CP |
| 1997-1998 | EP | EP | EP |
| 2002-2003 | CP | CP | CP |
| 2004-2005 | CP | CP | CP |
| 2006-2007 | CP | EP | / |
| 2009-2010 | CP | CP | CP |
| 2014-2015 | / | CP | / |
| 2015-2016 | EP | EP | EP |

**Table 1.** The classification results of EP and CP El Niño of the total 12 El Niño events from 1980 to 2017 using the Niño3/4 method and the EMI method.