# Peer review of "Impact of Eastern and Central Pacific El Niño on Lower"

_Atmospheric Chemistry and Physics, 2021_

## Author Comment (AC1)

Response to the review of "Impact of Eastern and Central Pacific El Niño on Lower Tropospheric Ozone in China":

We thank the referee for the detailed and constructive comments. We respond to each specific comment below. The referee's original comments are shown in blue. Our replies are shown in black. The corresponding changes in the manuscript are shown in *Italic black*.

**Anonymous Referee #1:**

Review of "Impact of Eastern and Central Pacific El Niño on Lower Tropospheric Ozone in China"

In this manuscript, the authors gave a very details study of middle to lower tropospheric ozone column (LTO) interannual variations induced individually by Eastern Pacific (EP) and Central Pacific (CP) El Niño. Due to the climatic factors (temperature, relative humidity, general circulations etc.) impact by CP&EP ENSO are varied much in regions of mid-latitude and uncertainties in extent, this kind of topic is of much difficulty to get solid conclusions. While this kind discussion is necessary. The analysis of the manuscript is mostly sound, but the scientific significance and basis of this study need emphasize and restate, and some details need clarify. My specific comments and suggestions listed below.

Specific comments:

1. I suggest to give an introduction on the relative accumulative variance (in percentage) that contribute to the climatic factors (T, RH, circulations etc.) impacted by CP&EP ENSO on the basis of previous studies. It's the scientific basis of this study.

Thanks for this suggestion. We have investigated previous studies on the impact of EP and CP ENSO on the climate factors in China. Most of them discussed precipitation, temperature, and water vapor transport, and they generally focused on the EP type and a specific variable. A few of them consider several variables in one study. For example, Li and Chen (2014) discussed the impact of ENSO on monthly means of daily maximum temperature (Tmax), minimum temperature (Tmin), precipitation, and relative humidity (RH). They found that South China will experience warmer temperatures, more precipitation, and higher humidity in El Niño years than in normal years. However, they did not distinguish between the different types of El Niño. These studies generally qualitatively describe the relationship between these variables and El Niño, so it's not easy to provide a quantitative evaluation of the relative accumulative variance on

different climate factors impacted by EP and CP ENSO. Moreover, their impacts vary with seasons and spatial location. As a result, we expanded our discussion in the introduction to provide a better basis for our research.

[Main text, Lines 69-83] :

The impact of ENSO on East Asia climate is known as the "Pacific-East Asia teleconnection", including the central Pacific cyclone, western North Pacific anticyclone, and the northeastern Asian cyclone (Wang et al., 2000; Zhang et al., 2011). During the developing autumn, the anomalous atmospheric circulation over the western North Pacific is nearly opposite in response to EP and CP El Niño. EP El Niño is generally accompanied by an anticyclone, while CP type usually has a cyclone over the western North Pacific. Yu and Sun (2018) found that East Asia winter monsoon is strong for EP ENSO but weak for CP ENSO. During the decaying phases of El Niño, Feng et al. (2011) showed that the EP type generally corresponds to the anomalous western Pacific anticyclone and brings ample moisture to southern China, contributing to the increased rainfall over these regions. However, the CP type generally has a weak western Pacific anticyclone and thus corresponds to the drier condition over southern China. Except for the rainfall patterns, other studies also show that different types of El Niño can induce different changes in tropical cyclone genesis and water vapor transport over China (Feng et al., 2011; Li et al., 2014; Wang and Wang, 2013). Accompanied by these meteorological changes, the two types of El Niño are also likely to exert different impacts on pollution conditions.

**2. In line 236, "over the west Pacific retreats"?**

Sorry for this lack of clarity. As this is not an important information, we deleted this sentence to avoid confusion for the readers. This will not affect the presentation of the main point in this section.

3. In line 303, I suggest "suppressed " replace "exceeded".

Thanks. We replaced it according to your suggestion.

4. In line 313, why is "chemistry still contributes positively over eastern China (Figure 6c)" oppositely corresponding to "reduction of SR in spring" in line 307.

In general, the reduction of solar radiation does contribute to the decrease of the chemical production of ozone. However, in our case, the temperature increase during the spring over most regions of eastern China (Figure 3o) counteracts the effect of SR reduction. We added the following discussion in the text:

[Main text, Lines 353-361] :

In EP condition, with the slight westward shift of the anticyclone center from winter to spring, the wind anomalies also shift from southwesterly to southerly, bringing more moisture, and further enhancing TP in higher latitudes where RH increases and SR decreases coherently. Although these changes are generally unfavorable to the local ozone production, the chemistry process still contributes positively to eastern China (Figure 6c). We attribute this pattern to the large-scale increase in temperature related to the warm south winds (Figure 30&s). As the climate warms from winter to spring, the role of temperature becomes increasingly important and may compensate or even exceed the impact of SR reduction.

5. In line 338, "the western Pacific warm pool begins to shrink with the building of La Niña (Johnson and Birnbaum, 2017)." is opposite to the reference in title "As El Niño builds, Pacific Warm Pool expands, ocean gains more heat".

This is because La Niña generally shows opposite characteristics to El Niño, according to Johnson and Birnbaum (2017). We revise the sentences here to avoid confusion.

[Main text, Lines 384-389] :

According to a previous study, the western tropical Pacific warm pool spreads eastward near the surface as El Niño builds (Johnson and Birnbaum, 2017). As La Niña generally shows opposite characteristics to El Niño, the western tropical Pacific warm pool under the former condition will shrink. Associated with the SST drop, SLP increases over the northwestern Pacific (Figure 4t), resulting in the enhanced western Pacific subtropical high(WPSH), a typical feature of CP El Niño (Chen et al., 2019).

6. In around line 343, related to ozone transport, please denote the upwind regions were in high ozone or low?

Thank you for pointing out this lack of clarity. We had a typo here (Figure 6g should be Figure 6p), and we now revised this sentence and also added an explanation.

[Main text, Lines 389-394] :

Controlled more by the local Pacific than the Indian Ocean, the SLP center shifts eastward during CP El Niño compared to the anomalous anticyclone during EP El Niño, and the positive LTO anomalies also move eastward accordingly (Figure S5h). Thus, the enhancement of LTO concentration in the SLP center cannot reach the southeast coastal line of China. Regional transport (Figure 6p) by the southwest wind anomalies surrounding the anomalous anticyclone (Figure 4t) brings air with low ozone content from the ocean and exerts a consistent negative contribution to LTO in southern China (Figure 2h; Jiang et al., 2021).

**References**

- Feng, J., Chen, W., Tam, C. Y. and Zhou, W.: Different impacts of El Niño and El Niño Modoki on China rainfall in the decaying phases, Int. J. Climatol., 31(14), 2091–2101, doi:10.1002/joc.2217, 2011.
- Jiang, Z., Li, J., Lu, X., Gong, C., Zhang, L. and Liao, H.: Impact of Western Pacific Subtropical High on Ozone Pollution over Eastern China, Atmos. Chem. Phys., 1–37, doi:10.5194/acp-2020-646, 2021.
- Li, Q. and Chen, J.: Teleconnection between ENSO and climate in South China, Stoch. Environ. Res. Risk Assess., 28(4), 927–941, doi:10.1007/s00477-013-0793-z, 2014.
- Li, X., Zhou, W., Chen, D., Li, C. and Song, J.: Water vapor transport and moisture budget over eastern China: Remote forcing from the two types of El Niño, J. Clim., 27(23), 8778–8792, doi:10.1175/JCLI-D-14-00049.1, 2014.
- Wang, B., Wu, R. and Fu, X.: Pacific-East Asian teleconnection: How does ENSO affect East Asian climate?, J. Clim., 13(9), 1517–1536, doi:10.1175/1520-0442(2000)013<1517:PEATHD>2.0.CO;2, 2000.
- Wang, C. and Wang, X.: Classifying el niño modoki I and II by different impacts on rainfall in southern China and typhoon tracks, J. Clim., 26(4), 1322–1338, doi:10.1175/JCLI-D-12-00107.1, 2013.
- Yu, S. and Sun, J.: Revisiting the relationship between El Niño-Southern Oscillation and the East Asian winter monsoon, Int. J. Climatol., 38(13), 4846–4859, doi:10.1002/joc.5702, 2018.
- Zhang, W., Jin, F. F., Li, J. and Ren, H. L.: Contrasting impacts of two-type El Niño over the western North Pacific during boreal autumn, J. Meteorol. Soc. Japan, 89(5), 563–569, doi:10.2151/jmsj.2011-510, 2011.

---

## Author Comment (AC2)

Response to the review of "Impact of Eastern and Central Pacific El Niño on Lower Tropospheric Ozone in China":

We thank the referee for the detailed and constructive comments. We respond to each specific comment below. The referee's original comments are shown in blue. Our replies are shown in black. The corresponding changes in the manuscript are shown in *Italic black*.

Anonymous Referee #2:

Review of "Impact of Eastern and Central Pacific El Niño on Lower Tropospheric Ozone in China"

This manuscript examines the effect of natural ENSO meteorological variability on lower tropospheric ozone over China. The authors use satellite data (IASI ozone retrievals) and GEOS-Chem model simulations, along with meteorological reanalysis data, to examine the effects of Eastern Pacific versus Central Pacific El Niños. The mechanisms responsible for the simulated (and observed) ozone changes are investigated by looking at the changes in meteorological variables (including solar radiation, relative humidity, temperature, sea-level pressure, and winds) and in ozone budget terms (primarily transport and chemistry, which are found to be the dominant drivers). This study extends past examinations of ENSO teleconnections to the middle to high latitudes, to examine effects on atmospheric chemical pollutants such as ozone. The study concludes that El Niño generally results in a decrease of lower tropospheric ozone over China, although with some regional and seasonal changes that differ between Eastern Pacific and Central Pacific El Niños. This work provides a useful new contribution to the literature examining ENSO teleconnections to the extratropics and is relevant to air pollution control policy in China. This paper would be suitable for publication in Atmospheric Chemistry and Physics with revisions to address concerns detailed below.

Major Comment

GEOS-Chem simulations: The study uses several sets of simulations with GEOS-Chem. First, a transient simulation is conducted for 1980-2017 with anthropogenic and biomass burning fixed at year-2000 levels, in order to assess the effects of ENSO-driven meteorological variability on ozone. The use of fixed emissions from biomass burning, which is known to exhibit large ENSO-driven variability, is a limitation of this study that should be discussed and justified more fully. (An additional set of simulations with interannually varying biomass burning emissions would add greatly to this study, but might be prohibitive for the authors to conduct at this stage.) A second set of simulations is

conducted using composite meteorological fields (compositing over 3 EP El Niño events, 4 CP El Niño events, and a full 30-year period). The use of composite meteorological fields, which will wash out most synoptic variability, seems like an odd choice, as opposed to simply compositing the results from the EP and CP El Niño years in the transient simulation. This approach needs to be justified and discussed more fully. Comment also on the implications of this meteorological compositing (versus compositing over events from full transient run) for issues of signal-to-noise in your results.

Thanks for the comments.

(1) The problem of biomass burning:

We admitted that the fixed biomass burning emission is a limitation of this study. An additional set of simulations with interannually varying biomass burning emissions would indeed help solve this problem. However, the biomass burning inventory is designated together with the fixation of anthropogenic emissions, and it's not easy to separate the two emissions at the moment. We consider it as a future work that requires more modeling skills.

We now added some discussions about how the inclusion of biomass variability could alter conclusions and a previous study on the effects of ENSO-modulated fires in section 4: Conclusion and discussion.

(2) Two sets of simulations:

In this study, we completed a historical run from 1980 to 2017 and a composite run driven by composite meteorological fields of EP, CP, and climatology conditions. We compared the results from the following two analyses:

1) Performing the composite analysis of simulation results from the historical run;
2) Driving the model with composite meteorological fields.

The reason why we conducted the second set of simulations is that the composite result from the historical run may contain perturbation by other climate signals, and because of the non-linear nature of ozone chemistry, the effect of these perturbations on ozone may be enlarged. So this analysis may be insufficient to prove that the composite difference is due solely to the ENSO-induced circulation and meteorological changes. Therefore, we conducted an additional set of simulations driven by composite meteorological fields to check the direct response of ozone to the ENSO-related meteorological changes. This treatment will indeed wash out most synoptic variability; however, as the focus of this study is to compare the difference between ENSO years and normal years on the interannual scale, the synoptic variability is the noise component rather than the signal.

Moreover, we found the two results from the two approaches largely agree (Figure R1 and Figure 2). In the main text, we decide to use the result of method 2) which contains a more direct ozone response (please see the response to P.8, l.169-174 below for more detailed revision).

[Figure]

**Figure R1.** The percentage changes (unit: %) of composite simulated (GC) tropospheric column ozone (0-6 km, unit: DU) anomalies for four seasons in EP and CP El Niño years from 1980-2017 historical simulation.

[Figure]

**Figure 2.** The percentage changes (unit: %) of simulated (GC) tropospheric column ozone (0-6 km, unit: DU) anomalies driven by composite meteorological fields for four seasons in EP and CP El Niño years. Areas with black dots indicate statistically significant changes.

Minor Comments

1. Introduction

Page 2, line 34 – Besides meteorological conditions, note that ozone concentrations are also largely controlled by precursor emissions (including anthropogenic emissions, which do not depend strongly on meteorology).

P.2, l.35 – Circulation and ventilation (i.e., transport) should also be listed as an important meteorological control on ozone.

We revised the sentences as below.

*[Main text, Lines 33-37]:*

*Tropospheric ozone concentration is largely affected by anthropogenic emissions, regional transport, and local meteorological conditions. Meteorological variables such as solar radiation, relative humidity, and temperature can influence the ozone precursor emissions and photochemical reaction rates* (Guenther et al., 2012; Jeong et al., 2018).

P.2, l.37 – Change "prominent interannual climate variabilities" to "prominent modes of interannual climate variability"

Revised.

P.3, l.65-69 – This is a sentence fragment. Please rewrite.

Thanks for pointing this out. We rearranged this paragraph.

*[Main text, Lines 63-69]:*

*A widely accepted view is to categorize El Niño into the Eastern Pacific (EP) and Central Pacific (CP) El Niño (Ashok et al., 2007; Yeh et al., 2009), whose positive sea surface temperature (SST) anomalies are located over the eastern and central Pacific respectively. Due to the different generation mechanisms (Yu et al., 2010) of the two types of El Niño, they can induce distinct changes in climate or synoptic weather in the tropics as well as the mid-to-high latitudes (Shi & Qian, 2018; Yu et al., 2012). ……*

2. Data and Methods

P.6, l.134 – Give months here. Should "Autumn" be "August"? Also, how do you start in September 2007, if as stated above, IASI-A started providing operational products in October 2007? (These dates are also mentioned at lines 163-164.)

It should be August; we revised this typo.

We added explanations in the manuscript to eliminate the confusion about IASI data time.

*[Main text, Lines 179-181]:*

*A 10-year (September 2007-August 2017) seasonal average was used as the climatological state. The missing month of IASI products data in September 2007 is filled as NaN in our calculation.*

 – Is this validation/evaluation done using the transient simulation with fixed year-2000 emissions? Would you expect results from such a simulation to match observed LTO values? You need to mention any caveats associated with this methodology here, and provide justification for why this approach was used.

This evaluation is done by using the transient simulation with fixed year-2000 emissions. We expected the IASI and GEOS-Chem simulation to show similar spatial patterns but may be different values. We have discussed the caveats with this method and provided potential reasons for the difference between GEOS-Chem and IASI *[Main text, Lines 237-260]*. We also added discussions about the reason why we use this approach.

*[Main text, Lines 172-187]* :

*The transient ozone simulation is further validated against tropospheric ozone within the same altitude range retrieved by the IASI. Because IASI only retrieves column ozone concentration between 0-6 km, our comparison and analysis also focus on 0-6 km integrated column ozone concentration, referred to as lower tropospheric ozone (LTO) thereafter. This focus on column ozone concentration can also reduce the impact of mismatch in anthropogenic emission between IASI and GEOS-Chem, which mainly influence the near-surface ozone concentration. As satellite observation starts in October 2007, to ensure comparability, we selected the 2015-2016 and 2009-2010 events to represent EP and CP El Niño, respectively. A 10-year (September 2007-August 2017) seasonal average was used as the climatological state. The missing month of IASI data in September 2007 is filled as NaN in our calculation. As we focus on the ozone changes, the bias induced by the mismatch of anthropogenic emissions is further mitigated by subtracting the climatological state. Therefore, we expected the ozone changes in ENSO years to show similar patterns during the ENSO years between GEOS-Chem simulation and IASI observation. Figure S1 shows the seasonal mean SST anomalies for the two periods selected, which correspond well to EP (2015-2016) and CP (2009-2010) El Niño patterns. The comparison results are discussed in Section 3.1 and Figure 1.*

 – As mentioned in Major Comment above, the approach of using composite meteorological fields needs to be explained more fully and justified here.

We now rewrote this paragraph and added more details.

*[Main text, Lines 189-199]* :

*To further distinguish the ozone changes between EP and CP El Niño, we also performed three composite model simulations driven by the composite meteorological fields of the four seasons of (1) the three most typical EP events (1982-1983, 1997-1998, 2015-2016), (2) the four most typical CP events (1994-1995, 2002-2003, 2004-2005, 2009-2010), and*

*(3) a 30-year averaged climatology (September 1985-August 2015). Figure S2 shows the composites of seasonal mean SST anomalies, which well corresponded to EP and CP El Niño. To save the computing resources and time, we calculated the seasonal mean and archived it in daily data files; each season is run for 10 days with the same seasonal-averaged meteorological fields every day. These three simulations started on the same day from the previous transient run to save the time for spin up. In this set of composite simulations, the difference between the result of simulations 1 and 3 (simulations 2 and 3) can represent the ozone changes driven by EP (CP) meteorological changes.*

Regarding the two different compositing approaches:

1) Performing the composite analysis of simulation results from the historical run (Figure R1);

2) Driving the model with composite meteorological fields and comparing the simulation result (Figure 2).

We added comparisons of these two methods in the supplementary and main text. The results from the two approaches are shown in Figures R1(merged into Figure S7) and Figure 2, respectively.

*[Supplementary, Text1]:*

*Text1. In this work, we applied two approaches to check the ozone response to El Niño:*

*(1) Performing the composite analysis of simulation results from the historical run (Figure S7a-h);*

*(2) Driving the model with composite meteorological fields and then comparing the simulation result (Figure 2).*

*The ozone changing patterns from these two approaches agree well for MAM and JJA, mostly agree for DJF, but showed some differences in southern China for SON under EP conditions. The SON difference is caused by the 1997-1998 El Niño. This event induced a comparable meteorology change with respect to other events but a much larger positive ozone change, which is opposite to other events (Figure S8). This might be a signal of model volatility and uncertainty. Since our purpose is to investigate how ozone responds to El Niño-induced meteorological changes, we think that driving the model using composite meteorological fields is more appropriate for our goal.*

[Figure]

**Figure S8.** The anomalies of simulated (GC) tropospheric column ozone ($O_3$), solar radiation (SR), relative humidity (RH), and temperature (T) during autumn ($SON_0$) for 1982-1983, 1997-1998, 2015-2016 EP El Niño events.

 *[Main text, Lines 264-268] :*

*Figure 2 shows the LTO changes in China during different seasons of the EP and CP El Niño. The patterns agree well with the composite results from historical simulations (Figure S7) but show stronger changing magnitudes due to the more direct response of ozone to meteorological changes. It is seen that LTO decreases over most regions in both EP and CP types in the range of 5~10% (2~5% in the composite of historical run),* ……

**3. Results**

P.9, l.187-188 – Add "for ozone" after "climatology state." Why are the seasons in Fig. S3 labeled with 0,1 subscripts. This is a climatology, not a composite of ENSO events, right?

Thanks. We revised the sentence accordingly and deleted the subscripts in FigS3.

P.9, l.193 – State reference period (Sep 2007-Aug 2017?) in Figure 1 caption.

We changed the Figure 1 caption from:

*Figure 1. The percentage changes (unit: %) of satellite-observed (IASI) and model-*

*simulated (GC) tropospheric column ozone (0-6 km, unit: DU) for four seasons in EP (2015-2016) and CP (2009-2010) El Niño years.*

to:

*Figure 1. The percentage changes (unit: %) relative to climatology state (Sep. 2007-Aug. 2007) of satellite-observed (IASI) and model-simulated (GC) tropospheric column ozone (0-6 km, unit: DU) for the four seasons in EP (2015-2016) and CP (2009-2010) El Niño years.*

P.10, l.219-220 – Change to "poor *representation* of *the* Brewer-Dobson circulation." Explain how the B-D circulation influences lower tropospheric ozone here. Is there evidence that the distribution of ozone in the stratosphere is biased? Or, do you just mean here that the stratospheric influence on LTO is poorly represented (e.g., from biases in stratosphere-troposphere exchange, or high-lat to mid-lat mixing in the troposphere)?

We added explanations about how BDC influences lower tropospheric ozone. Here we just mean the bias due to stratosphere-troposphere exchange. Currently, the magnitude of the stratospheric contribution and its importance in the tropospheric ozone budget are poorly constrained (Neu et al., 2014). In the "GEOS-Chem Steering Committee Telecon, December 17, 2013 10-11:30 Eastern" it is mentioned that the rate of BDC may be underestimated in GEOS-Chem:

"GEOS-FP O3 columns also much lower from OMI columns currently being used in GEOS-Chem. Brewer-Dobson circulation is a little more sluggish in GEOS-FP so would expect to be lower, but Dylan Jones surprised at the magnitude, considering that O3 in GEOS-FP is assimilated."

We also added more details about it in the text.

*[Main text, Lines 244-257] :*

*Another reason is that the model underestimates the average ozone concentration at high latitudes in winter and spring (Figure S3), which leads to less ozone transport from polar regions to northern China in the model. The IASI-retrieved data exhibits high ozone concentration in the Arctic during winter and spring (Figure S3f, g); this phenomenon is also shown in previous studies (Cooper et al., 2014). However, the GEOS-Chem simulation did not capture the high values in polar regions. A possible explanation for this underestimation is that the Brewer-Dobson circulation may be insufficiently represented in the model. BDC consists of an upward transport branch across the tropopause in the tropics and has a strong poleward and downward circulation branch in the winter hemisphere (Hu et al., 2017), which contributes to the high LTO concentration in polar regions through the stratosphere-troposphere exchange. Another potential reason for the underestimation is due to the unprecise halogen chemistry in GEOS-Chem. Wang et al. (2021) point out that the halogen chemistry can worsen the underestimation of*

*tropospheric ozone in the Northern Hemisphere by halogen-catalyzed loss.*

P.12, l.260 – "insufficient" compared to what? Perhaps change wording to "negligible."

Thanks. We changed the word accordingly.

P.12, l.265, Figure 5 – Explain what is being plotted here. This figure is quite confusing. Is the absolute value of the tendency due to each process taken in each grid box, or after the full field is summed over the study domain?

Thanks for pointing out this confusion. We added more explanation about this figure.

*[Main text, Lines 298-305]:*

*In addition, we calculate the budget changes corresponding to the EP and CP events from GEOS-Chem simulations. The simulated ozone concentration is determined mainly by four processes, including chemistry, transport, mixing, and convection. Since each process can contribute to ozone either positively or negatively, we calculated the absolute value of the column integrated ozone budget in each grid box and then calculated the mean value of the chosen domain (24.0–42.0°N, 100.0–117.5°E, purple box in Figure 6a) to better quantify the impact of each process. The results are shown in Figure 5.*

P.12, l.266, Figure 6 – Confusing. Does this figure show a composite of tendencies due to these processes, or just the values from a single simulation with composite meteorology? Clarify in figure caption.

Thanks for pointing out this confusion. This is the result of a single simulation with composite meteorology. We clarified it in the figure caption and added more explanations in the text.

*[Figure 6 caption]:*

*Figure 6. The tropospheric column ozone mass anomalies of chemistry and transport processes (0-6km, unit: kg d$^{-1}$) driven by composite meteorological fields for four seasons in EP and CP El Niño years.*

*[Main text, Lines 307-308]:*

*Figure 6 shows the spatial distribution of ozone budgets corresponding to the chemistry and transport processes from the simulation driven by composite meteorological fields.*

P.12, l.270 – "Southwestern" → "Southwesterly"

Replaced.

Replaced.

We revised the sentences as below.

*[Main text, Lines 333-336]* :

*WNPAC is a critical system that links El Niño and East Asia climate change, and its formation and maintenance mechanisms are discussed thoroughly in Li et al. (2017). WNPAC is initiated and maintained by local atmosphere-ocean interaction (Wang et al., 2000) and the moist enthalpy advection/Rossby wave modulation (Wu et al., 2017a, 2017b).*

Replaced.

P.16, l.343-345 – This manuscript uses acronyms very heavily. This is generally fine as a convenient shorthand–but occasionally, as in this sentence, it makes it very difficult for a reader to follow: "Controlled more by local Pacific than IO, the SLP center shifts eastward compared to AAC in EP, and the positive LTO anomalies also move eastward accordingly." Try to rewrite.

Thanks for pointing this out. We revised the sentences as below. We also reduced the use of acronyms in other places.

*[Main text, Lines 389-392]* :

*Controlled more by the local Pacific than the Indian Ocean, the SLP anomaly center shifts eastward during CP El Niño compared to the anomalous anticyclone during EP El Niño, and the positive LTO anomalies also move eastward accordingly (Figure S5h).*

4. Conclusions and discussion

P.16, l.362 – Write out "western North Pacific anomalous anticyclone" on its first use in this section to aid readers looking only at the Conclusions.

We added the full name of WNPAC (western North Pacific anomalous anticyclone) in its first appearance in this section (line 410).

P.17, l.365-368 – Run-on sentence. Please split into two sentences or rephrase.

We revised the sentences as below.

*[Main text, Lines 413-417] :*

*In spring, the WNPAC persisted under EP conditions and kept impacting LTO; thus, the regional transport dominates the overall decline of LTO by 5~10%. However, the role of transport weakens due to the disappearance of WNPAC under CP conditions. On the other hand, the local ozone production increases due to the drier environment, which leads to a slight ozone increase (+0~4%) over southern China.*

P.17, l.371 – Write out "western pacific subtropical high" on first use in this section.

We replaced it with the full name.

P.17, l.389-390 – As mentioned in Major Comment above, the omission of interannually varying biomass burning from this study is a significant issue in assessing the effects of ENSO on LTO over China. You should elaborate on how inclusion of biomass variability could alter conclusions of this study, and comment on previous studies on the effects of this variability (if any)

Thanks for pointing out this shortage. We added discussions in this section.

*[Main text, Lines 438-446] :*

*The variation of biomass burning emission is not included in our study. However, the increased frequency and intensity of wildfires induced by El Niño over Southeast Asia and Australia can generate more carbon monoxide, which is an important ozone precursor. The LTO changes should be even larger than the simulated results shown in this study. A previous study shows that the ENSO-modulated fires in Southeast Asia dominate the subtropical trans-Pacific ozone transport during the springtime (Xue et al., 2021). Based on the structure of the wind fields (Figure 3q-t, 4q-t), the impact of long-distance transportation from Southeast Asia to China is relatively small, and thus its impact on the spatial patterns of LTO changes in China is limited. The role of biomass burning emissions on ozone will be quantitatively investigated in the future.*

**References**

Feng, J., Chen, W., Tam, C. Y. and Zhou, W.: Different impacts of El Niño and El Niño Modoki on China rainfall in the decaying phases, Int. J. Climatol., 31(14), 2091–2101, doi:10.1002/joc.2217, 2011.

Guenther, A. B., Jiang, X., Heald, C. L., Sakulyanontvittaya, T., Duhl, T., Emmons, L. K. and Wang, X.: The model of emissions of gases and aerosols from nature version 2.1 (MEGAN2.1): An extended and updated framework for modeling biogenic emissions, Geosci. Model Dev., 5(6), 1471–1492, doi:10.5194/gmd-5-1471-2012, 2012.

Hu, D., Guo, Y., Wang, F., Xu, Q., Li, Y., Sang, W., Wang, X. and Liu, M.: Brewer-Dobson Circulation: Recent-Past and Near-Future Trends Simulated by Chemistry-Climate Models, Adv. Meteorol., 2017, 18–20, doi:10.1155/2017/2913895, 2017.

Jeong, J. I., Park, R. J. and Yeh, S. W.: Dissimilar effects of two El Niño types on PM2.5 concentrations in East Asia, Environ. Pollut., 242, 1395–1403, doi:10.1016/j.envpol.2018.08.031, 2018.

Koumoutsaris, S., Bey, I., Generoso, S. and Thouret, V.: Influence of El Niño-Southern Oscillation on the interannual variability of tropospheric ozone in the northern midlatitudes, J. Geophys. Res. Atmos., 113(19), 1–21, doi:10.1029/2007JD009753, 2008.

Li, T., Wang, B., Wu, B., Zhou, T., Chang, C. P. and Zhang, R.: Theories on formation of an anomalous anticyclone in western North Pacific during El Niño: A review, J. Meteorol. Res., 31(6), 987–1006, doi:10.1007/s13351-017-7147-6, 2017.

Li, X., Zhou, W., Chen, D., Li, C. and Song, J.: Water vapor transport and moisture budget over eastern China: Remote forcing from the two types of El Niño, J. Clim., 27(23), 8778–8792, doi:10.1175/JCLI-D-14-00049.1, 2014.

Neu, J. L., Flury, T., Manney, G. L., Santee, M. L., Livesey, N. J. and Worden, J.: Tropospheric ozone variations governed by changes in stratospheric circulation, Nat. Geosci., 7(5), 340–344, doi:10.1038/ngeo2138, 2014.

Olsen, M. A., Wargan, K. and Pawson, S.: Tropospheric column ozone response to ENSO in GEOS-5 assimilation of OMI and MLS ozone data, Atmos. Chem. Phys., 16(11), 7091–7103, doi:10.5194/acp-16-7091-2016, 2016.

Oman, L. D., Douglass, A. R., Ziemke, J. R., Rodriguez, J. M., Waugh, D. W. and Nielsen, J. E.: The ozone response to enso in aura satellite measurements and a chemistry-climate simulation, J. Geophys. Res. Atmos., 118(2), 965–976, doi:10.1029/2012JD018546, 2013.

Shi, J. and Qian, W.: Asymmetry of two types of ENSO in the transition between the East Asian winter monsoon and the ensuing summer monsoon, Clim. Dyn., 51(9–10), 3907–3926, doi:10.1007/s00382-018-4119-1, 2018.

Wang, B., Wu, R. and Fu, X.: Pacific-East Asian teleconnection: How does ENSO affect East Asian climate?, J. Clim., 13(9), 1517–1536, doi:10.1175/1520-0442(2000)013<1517:PEATHD>2.0.CO;2, 2000.

Wang, C. and Wang, X.: Classifying el niño modoki I and II by different impacts on rainfall in southern China and typhoon tracks, J. Clim., 26(4), 1322–1338, doi:10.1175/JCLI-D-12-00107.1, 2013.

Wang, X., Jacob, D. J., Downs, W., Zhai, S., Zhu, L., Shah, V., Christopher, D., Alexander, B., Evans, M. J., Eastham, S. D., Andrew, J., Veres, P., Koenig, T. K., Volkamer, R., Huey, L. G., Thomas, J., Percival, C. J., Lee, B. H. and Thornton, J. A.: Global tropospheric halogen ( Cl , Br , I ) chemistry and its impact on oxidants, Atmos. Chem. Phys., (June), 1–34, 2021.

Wu, B., Zhou, T. and Li, T.: Atmospheric dynamic and thermodynamic processes driving the western North Pacific anomalous anticyclone during El Niño. Part I: Maintenance mechanisms, J. Clim., 30(23), 9621–9635, doi:10.1175/JCLI-D-16-0489.1, 2017a.

Wu, B., Zhou, T. and Li, T.: Atmospheric dynamic and thermodynamic processes driving the western north Pacific anomalous anticyclone during El Niño. Part II: Formation processes, J. Clim., 30(23), 9637–9650, doi:10.1175/JCLI-D-16-0495.1, 2017b.

Xu, L., Yu, J. Y., Schnell, J. L. and Prather, M. J.: The seasonality and geographic dependence of ENSO impacts on U.S. surface ozone variability, Geophys. Res. Lett., 44(7), 3420–3428, doi:10.1002/2017GL073044, 2017.

Xue, L., Ding, A., Cooper, O., Huang, X., Wang, W., Zhou, D., Wu, Z., McClure-Begley, A., Petropavlovskikh, I., Andreae, M. O. and Fu, C.: ENSO and Southeast Asian biomass burning modulate subtropical trans-Pacific ozone transport, Natl. Sci. Rev., 8(6), doi:10.1093/nsr/nwaa132, 2021.

Yu, J. Y., Kao, H. Y. and Lee, T.: Subtropics-related interannual sea surface temperature variability in the central equatorial pacific, J. Clim., 23(11), 2869–2884, doi:10.1175/2010JCLI3171.1, 2010.

Yu, J. Y., Zou, Y., Kim, S. T. and Lee, T.: The changing impact of El Nio on US winter temperatures, Geophys. Res. Lett., 39(15), doi:10.1029/2012GL052483, 2012.

Ziemke, J. R. and Chandra, S.: La Nina and El Nino - Induced variabilities of ozone in the tropical lower atmosphere during 1970-2001, Geophys. Res. Lett., 30(3), 30–33, doi:10.1029/2002GL016387, 2003.